# The Impact of Living Arrangements on the Prevalence of Falls after Total Joint Arthroplasty: A Comparison between Institutionalized and General Geriatric Population

**DOI:** 10.3390/ijerph20043409

**Published:** 2023-02-15

**Authors:** Anca Maria Pop, Octav Marius Russu, Sándor György Zuh, Andrei Marian Feier, Tudor Sorin Pop

**Affiliations:** 1Faculty of Medicine, George Emil Palade University of Medicine, Pharmacy, Science, and Technology of Târgu Mureș, 540139 Târgu Mureș, Romania; 2Department of Orthopedics and Traumatology, George Emil Palade University of Medicine, Pharmacy, Science, and Technology of Târgu Mureș, 540139 Târgu Mureș, Romania; 3Clinic of Orthopedics and Traumatology, County Clinical Hospital Mureș, 540080 Târgu Mureș, Romania

**Keywords:** total joint replacement, elderly, accidental falls, institutionalized persons

## Abstract

Due to population aging, there is an increasing need for orthopedic surgery, especially total knee arthroplasty (TKA) and total hip arthroplasty (THA). In geriatric patients, postoperative falls are common events which can compromise the success of these expensive procedures. The aim of our study was to assess the influence of living arrangements on the prevalence of postoperative falls following joint replacement. We included 441 patients after TKA or THA, living in nursing homes, alone or with family. The prevalence of falls in the first 2 years (15.2%) was significantly influenced by living arrangements: patients with TKA or THA living alone had three times higher odds of falling compared to those living with family, and institutionalized patients with THA had four times higher odds of falling compared to those living with family. Of 67 patients who fell, 6 (8.9%) needed reintervention. For TKA patients, the fall rates were not significantly different between institutions and family, indicating the interest of nursing homes in offering proper care. However, for the THA group, the results were poorer, emphasizing the need for improvement in postoperative rehabilitation. Further multi-centric studies are required for generalizing the impact of living arrangements on fall prevalence after joint replacement.

## 1. Introduction

The increasing number of elderly people is a globally encountered phenomenon, progressing from 10% of adults aged over 60 years in 2000 and estimated to exceed 20% by 2030, which also leads to a higher proportion of geriatric patients requiring orthopedic surgery [1]. Common surgical procedures in this category of patients are total knee arthroplasty (TKA) for severe knee osteoarthritis (OA), total hip arthroplasty (THA) for both end-stage hip OA (90% of the cases) and displaced femoral neck fractures (5% of the cases) [2,3].

OA is the most frequent joint disease [4] and a major cause of disability worldwide, with increasing incidence especially in older patients [5,6,7]. It mainly affects the hip and the knee and was considered the leading source of limitation in daily activities such as walking, dressing, or object carrying, as reported based on the French “Disability-Health survey” [8]. In the context of continuous aging due to increased life expectancy and sedentary lifestyle leading to obesity, OA and diabetes were the most important pathologies contributing to the rise in the number of years lived with disability by comparing the time frame of 1990–2005 with 2005–2015 [9,10]. Women are more frequently affected compared to men and also tend to present with more severe radiographic features of hip and knee OA [11]. Aging is one of the most remarkable risk factors for OA development, as the highest prevalence of this disease was estimated in patients over 70 years [9,12,13]. Apart from its impact on physical health by increasing cardiovascular risk [14], OA can also have consequences regarding patients’ psychological health [6]. For example, data from the literature suggested a positive correlation between knee OA and the development of depressive symptoms, anxiety, reduced sleep quality and even suicidal ideation in middle-aged and elderly patients [15]. Moreover, patients with hip OA also had more intense depressive symptoms compared to those with other chronic conditions such as cancer, diabetes or cardiovascular disease [16]. In elderly people, the impairment of physical and mental health due to OA is an important risk factor for falls, as nearly 50% of those with OA report falls every year [17]. The impact of falls becomes more obvious due to population aging but also due to the consistent health costs and possible severe consequences such as head trauma or fractures which can be associated with it [18,19]. 

Total joint arthroplasty (TJA) is the most effective but also expensive treatment for both knee and hip OA by providing improved joint mobility and pain reduction [20]. The prevalence of falls after surgical intervention remains high, as almost one-third of patients report at least one fall in the first year after THA or TKA. The growing need for TJA, illustrated by the estimated increases of 171% for THA and 185% for TKA until 2030 [21,22], warrants the investigation of conditions associated with postoperative falls in this category of patients. Until now, several risk factors have been identified, such as persistent disability and pain after TJA [23,24,25], deficient limb proprioception [25], depression, a history of preoperative falls [26], female gender [27], older age [28], medication (bisphosphonates and drugs used for psychiatric diseases) [29,30] and age-related neuromuscular alteration [28]. 

Some of the aforementioned conditions are characteristics of geriatric syndromes and are often incriminated as predictors of the future need for institutionalization [31]. Although older people today usually have better functional capabilities than their predecessors, the need for institutional care still remains high for old people with significant cognitive impairments and a history of falls [31,32]. Scientific data suggest that institutionalization has a negative impact on the quality of life of elderly people, due to poor social activity, a sedentary lifestyle, a lack of preoccupation with physical health, an association of chronic diseases [33] and a high prevalence of depression [34]. Additionally, elderly people living in nursing homes have higher odds of falling compared to the community-dwelling ones [35]; this exposes them to an increased risk of severe injury such as femoral neck fractures requiring THA or periprosthetic fractures in the case of those who already benefited from TKA or THA due to OA. 

As there is limited information regarding the impact of living arrangements on the risk of falls after TJA, the aim of our study was to evaluate the prevalence of falls in the first two years following joint replacement surgery in elderly people living in nursing homes (institutions), living alone and living in a family environment. Moreover, among reported falls, we aimed to identify the percentage which required surgical intervention. The null hypothesis to be tested was that living arrangements do not influence the rate of postoperative falls following TJA. 

## 2. Materials and Methods

### 2.1. Study Design and Participants

This retrospective study was conducted on 441 patients who underwent TKA or THA between 2016 and 2020 in the Clinic of Orthopedics and Traumatology of the County Clinical Hospital Mureș, followed up for two years after surgery and divided into three groups according to their living arrangements: institutionalized patients (living in nursing homes), patients living alone in the community and patients living in the community within the family. As inclusion criteria, we selected patients aged over 70 years who had an indication for TJA following trauma or advanced hip/knee OA. We excluded patients who at the time of surgery had been previously diagnosed with significant cognitive impairment, patients with walking disabilities after stroke or patients who did not show up to regular follow-up controls. The following variables were retrieved from the medical registers: age, gender, associated comorbidities, history of falls prior to surgery, falls reported in the next two years after TKA or THA and number of re-interventions after falls. As part of the clinical protocol, the patients are generally instructed to report postoperative falls and are also specifically asked about postoperative falls during the usual follow-up controls at 3 months, 1 year and 2 years after TJA. 

Based on the variable fall rates after TJA reported in the scientific literature, which range between 6 and 42% [17,28], we chose a conventional prevalence of 50% in order to estimate the necessary sample size for our study. This was calculated based on the following formula: $n=\frac{Z^{2}\times P\left(1-P\right)}{d^{2}}$, considering n = the sample size, Z = the statistic value of 1.96 for a level of confidence of 95%, P = the chosen expected prevalence of 50% and d = the precision (according to the effect size) with a value of 0.05, which is considered appropriate for a prevalence between 10 and 90% [36]. The minimum sample size recommended for this study was 384. 

### 2.2. Statistical Analysis 

Statistical analysis was carried out using GraphPad Prism 7 for Windows (GraphPad Software, San Diego, CA, USA). Categorical data were assessed by Chi-square test and Fisher’s exact test with Bonferroni correction for post hoc analysis, whereas continuous data were evaluated using the Kruskal–Wallis test. In order to assess the contribution of multiple variables to the occurrence of postoperative falls, we conducted multiple logistic regression. The continuous variables were expressed as mean ± standard deviation (SD) and categorical variables as percentages and frequency distribution. The level of statistical significance was set at a *p* value < 0.05 (two-tailed).

## 3. Results

The clinical and demographic data of the patients included in our study are presented in Table 1. 

Regarding gender, the type of surgical intervention and preoperative history of falls, the three study groups according to living arrangements did not show significant statistical differences. The only parameter which differed significantly among the three groups was age, as institutionalized patients were significantly older compared to those living in the community alone or with family.

The prevalence of falls after TKA and THA according to living arrangements is presented in Table 2. The presence of falls was reported in 36 patients (20.3%) out of 177 after TKA and in 31 patients (11.7%) out of 264 after THA. There was a statistically significant difference between the fall rates recorded in patients according to their living arrangements in both types of arthroplasty (*p* = 0.009 for TKA and *p* = 0.017 for THA) (Table 2). 

In order to evaluate the differences between individual groups, we conducted the post hoc analysis using Fisher’s exact test with Bonferroni correction and adjusted the level of statistical significance at a value of *p* < 0.0167 (Table 3). 

Patients living alone had statistically significantly higher odds of falling after both TKA (odds ratio (OR): 3.632, *p* = 0.006) and THA (OR: 3.529, *p* = 0.013). Institutionalized patients had higher odds of falling after both TKA (OR: 3.086, *p* = 0.03) and THA (OR: 4.653, *p* = 0.003), but the difference was statistically significant only in the case of THA. Moreover, compared to patients living alone, the institutionalized ones had lower odds of falling after TKA (OR: 0.849, *p* = 0.817) and higher after THA (OR: 1.319, *p* = 0.656), but without statistical significance in any of the cases. 

In order to assess the contribution of the other demographic and clinical variables to the occurrence of postoperative falls, we conducted multivariate logistic regression as presented in Table 4. 

The strongest contributors to the presence of postoperative falls were living alone and polypharmacy (*p* < 0.0001), followed by increasing age (*p* = 0.004) and a history of preoperative falls (*p* = 0.02). 

Out of the 67 patients who reported falls in the first 24 months after surgery, 6 (8.9%) required revision, four after THA and two after TKA, respectively. 

## 4. Discussion

TKA and THA are cost-effective treatment strategies for knee and hip end-stage OA, by restoring joint mobility, relieving pain and increasing patients’ quality of life [37]. The surgical management of displaced femoral neck fractures is still under debate, but current data recommend the use of THA instead of hemiarthroplasty in older patients (over 65 years) without cognitive or walking impairment [38]; THA was associated with better hip function and quality of life, along with a lower need for re-intervention, estimated at 4% vs. 7% after hemiarthroplasty [39]. Based on these considerations, the protocol of our clinic implies the use of THA as the treatment of choice in elderly patients with displaced femoral neck fractures who comply with the aforementioned indications; therefore, patients with cognitive deficiencies or reduced mobility secondary to stroke were not included in our analysis, as these are better candidates for hemiarthroplasty [38]. 

In our study, we found a fall prevalence of 15.2% after TJA, which is in accordance with previously published scientific data, reporting values ranging between 3.1–51.8% in the first two years following TKA or THA [28]. 

Regarding TKA, several authors have reported a prevalence ranging between 6.23 and 32.9% in the first year after surgery: Matsumoto et al. [40] found a prevalence of 32.9%, Tsonga et al. [41] calculated a prevalence of 22.1% and Si et al. [27] estimated a prevalence of 6.23%. After a follow-up period of two years, we found a prevalence of 20.3%, which resembles the previously published data. The prevalence does not proportionally increase with the time elapsed after surgery, as in a cohort of 376 participants, Chan et al. [25] observed that 17.2% of patients fell at a median time of 15 weeks after TKA, whereas Matsumoto et al. [42] concluded that after up to 6 years of follow-up, the fall rate was 38.9%. Furthermore, Si et al. [27] identified that the most important associated risk factors were age over 70 years, advanced OA of the contralateral knee and female gender. Our results are similar to those of Tsonga et al. [41], who also mentioned that the preoperative history of falls and the fear of falling increase the risk of this event by seven and twelve times, respectively. Periprosthetic fractures are a serious but not very common complication after TKA, accounting for approximately 3.6% of revisions [43]. In our study group, two patients (5.6%) out of thirty-six who reported falls developed supracondylar femoral fractures, which is recognized as the most common type of periprosthetic fracture, involving low-velocity trauma [44]. 

Regarding THA, we found a fall prevalence of 11.7% in the first 24 months after surgery, which is lower than data from other studies that reported fall rates in the first 12 months between 25 and 36% [29,45,46]. The risk of fractures after falling ranges between 5 and 14% [29,45,46]; therefore, patients with THA are considered to be at an increased risk of fall-induced lesions, such as periprosthetic fractures and joint dislocation [46]. In our study, from thirty-one patients with falls after THA, four (12.9%) developed periprosthetic fractures which required surgical intervention. 

We found a statistically significant difference between the fall rates of patients with TKA and THA according to their living arrangements. Patients with TKA living alone had 3.632 times higher odds of falling (*p* = 0.006) compared to those living with their family, but slightly lower odds (0.849) compared to those living in institutions, without statistical significance (*p* = 0.817), however. Interpreted separately, institutionalized patients with TKA tended to fall significantly more than those living with family, but in the context of a larger contingency table, the results did not reach the threshold of statistical significance (*p* = 0.03, higher than the value of *p* < 0.0167 considered significant after applying the Bonferroni correction). Patients with THA living in institutions and alone had significantly higher odds of 4.653 and 3.529, respectively, of falling compared to those living with family; however, no difference in the fall rate was found between those living in nursing homes or alone.

Older people living alone are considered a vulnerable group prone to an increased risk of falling with possible severe consequences [47,48]. They tend to be less involved in activities, causing a reduction in mobility, and experience an increased fear of falling [48]. Jørgensen and Kehlet [30] reported that patients with TKA or THA living alone had a double risk of postoperative falls compared to patients living with others. Moreover, in a cohort of patients with THA, Edwards et al. [49] found that compared to patients living with others, those living alone had a higher incidence of revision due to fracture, infection or implant dislocation (2.2%) and also a higher incidence of mortality (3%). These complications were observed more frequently in the first 90 days after THA, when special care is required for daily activities in order to diminish the risk of falling [49]. Additionally, after adjusted multivariate analysis, Couderc et al. [50] concluded that living alone was the only variable correlated with postoperative complications arising in the first 90 days after THA or TKA (adjusted OR = 3.2). Our findings are in accordance with the aforementioned data, as based on the multivariate analysis, living alone was considered the strongest predictor for postoperative falls (*t*-value 5.43, *p* < 0.0001).

In our study, the fall rate encountered in institutionalized patients compared to those living alone was lower after TKA (27% vs. 30.3%) and higher in the case of THA (19.7% vs. 15.7%), without reaching statistical significance, however. Falls in nursing homes are frequent events [51], associated with several risk factors, such as older age, the use of medication (antidepressants, sedatives or antipsychotics) and moderate disability rather than severe frailty when people need help from a caregiver for walking [52]. Our data showed that there is no statistically significant difference regarding the prevalence of falls between people living in institutions and alone, which is in accordance with the results published by Datta et al. [53], who concluded that environmental factors (in this case, the living arrangements) can be the cause of more than 30% of falls. The high prevalence of falls recorded in nursing institutions warrants the development of preventive strategies [54]; however, there is a lack of scientific evidence presenting efficient methods for lowering the incidence of this phenomenon [55].

### Strengths and Limitations of the Study 

To the best of our knowledge, this is the only study evaluating the influence of three types of living arrangements (living alone, in nursing homes and with family) on the prevalence of falls after TJA. Most previous studies evaluated only institutionalized patients or patients living alone compared to those living with others, without distinguishing between family and nursing homes. In our country, we face two important aspects: the increasing tendency of placing elderly people in nursing homes and the free access to TJA due to a national health program for joint replacement. Therefore, the increasing number of TJA interventions calls for the development of preventive measures and better rehabilitation protocols in order to maintain the high success rate of this expensive procedure. On the other hand, our results cannot be generalized as they were based exclusively on specific conditions offered by Romanian nursing homes, and their organization and available care facilities may differ significantly from other countries. It is important to mention that in Romania, patients do not receive physical therapy in the community or nursing homes, as this is only provided during hospitalization following surgical intervention.

Moreover, we did not conduct a separate analysis for patients with and without a preoperative history of falls, which is known to be a predictor of postoperative falling. Another limitation is the collection of data based on medical registers, as postoperative falls are not always properly reported by patients or their caregivers to medical authorities. Furthermore, the design of the study does not allow the confirmation of causality between the studied variables, but only a possible association between them. Larger multi-centric, prospective and randomized studies are needed for the more accurate evaluation of the impact of living arrangements on the fall rates after TJA in elderly patients. 

## 5. Conclusions

Elderly patients living alone or in institutions have similar odds of falling after joint replacement surgery. The family offers the safest environment, as these patients encountered significantly lower odds of falling compared to those living alone. The fall rates of patients with TKA living with family were comparable to those living in institutions, encouraging us to conclude that nursing homes make efforts in ensuring proper care. However, these efforts must be improved for elderly people with THA, as these faced significantly higher fall rates and the need for revision interventions.

## Figures and Tables

**Table 1 ijerph-20-03409-t001:** Characteristics of the study group.

Variables		Whole Study Group	Living Arrangements			
			Institutionalized Patients	Living Alone	Living with the Family	*p* Value
Age (Mean ± SD) (Years)		77.1 ± 4.5	80.8 ± 3.2	75.7 ± 4.3	77.8 ± 4.2	<0.0001 *
Gender	Female	248 (56.2%)	46	78	124	0.13 **
	Male	193 (43.8%)	50	61	82	
	Total	441 (100%)	96	139	206	
Type of intervention	THA (secondary to femoral neck fractures)	78 (17.7%)	17	23	38	0.87 **
	THA (secondary to hip OA)	186 (42.2%)	44	60	82	
	TKA (secondary to knee OA)	177 (40.1%)	35	56	86	
History of falls prior to surgery	Present	182 (41.3%)	38	58	86	0.14 **
	Absent	259 (58.7%)	58	81	120	
Associated comorbidities/geriatric syndromes	Frailty syndrome	57 (12.9%)	27	8	22	
	Depression	108 (24.5%)	26	47	45	-
	Malnutrition	42 (9.5%)	24	8	10	
	Obesity	82 (18.6%)	12	31	39	
	Parkinson disease	36 (8.2%)	12	5	19	
	Polypharmacy	232 (52.6%)	52	58	122	

* *p* value obtained based on Kruskal–Wallis test, ** *p* value obtained based on Chi-square test. TKA = total knee arthroplasty, THA = total hip arthroplasty, OA = osteoarthritis.

**Table 2 ijerph-20-03409-t002:** The distribution of falls after TKA and THA according to living arrangements.

Living Arrangements	History of Falls after TKA		History of Falls after THA	
	Falls Present	Falls Absent	Falls Present	Falls Absent
Institutionalized	10 *	27	12 **	49
Living alone	17 *	39	13 **	70
Living with the family	9 *	75	6 **	114
*p* value	0.009 *		0.017 **	

*^,^** statistically significant difference based on Chi-square test. TKA = total knee arthroplasty, THA = total hip arthroplasty.

**Table 3 ijerph-20-03409-t003:** Results of the post hoc analysis based on Fisher’s exact test.

Comparison between Groups	History of Falls after TKA		History of Falls after THA	
	Odds Ratio (95% CI)	*p* Value	Odds Ratio (95% CI)	*p* Value
Institutionalized and Living alone	0.849 (0.337–2.138)	0.817	1.319 (0.554–3.134)	0.656
Institutionalized and Living with the family	3.086 (1.113–8.411)	0.03	4.653 (1.651–13.11)	0.003 *
Living alone and Living with the family	3.632 (1.483–8.899)	0.006 *	3.529 (1.282–9.711)	0.013 *

* statistically significant difference based on Fisher’s exact test with Bonferroni correction and adjusted *p* < 0.0167. TKA = total knee arthroplasty, THA = total hip arthroplasty.

**Table 4 ijerph-20-03409-t004:** Results of the multivariate logistic regression.

Independent Variables	*t*-Value	*p* Value
Age	2.86	0.004 *
Gender	0.15	0.87
Living in nursing institutions	1.05	0.29
Living alone	5.43	<0.0001 *
THA	1.215	0.22
Frailty	1.55	0.12
Depression	1.45	0.14
Malnutrition	0.38	0.7
Obesity	0.52	0.6
Parkinson disease	1.78	0.07
Polypharmacy	4.86	<0.0001 *
History of preoperative falls	2.18	0.02 *

* statistically significant difference.

## Data Availability

All relevant data are contained within the article.

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
