# Peer review of "The Impact of Living Arrangements on the Prevalence of Falls after Total Joint Arthroplasty: A Comparison between Institutionalized and General Geriatric Population"

_ijerph, 2023, doi:10.3390/ijerph20043409_

Round 1
Reviewer 1 Report
I have completed my review of the manuscript titled "The impact of living conditions on the prevalence of falls after total joint arthroplasty: A comparison between institutionalized and general geriatric population"
Introduction
The research hypotheses are missing. The aim needs to be more precise. What exactly is the goal? Assessment of fall, re-fall, or need for surgical intervention…?
"The aim of our study was to evaluate the prevalence of falls and the rate of reintervention in the first two years"
In my opinion, the results only indicate an assessment of a further decline, not a required surgical revision. Clearly describe it in the goal, because the surgical revision is visible only in the last sentence of the paragraph results and in the pictures, that does not mean the result. The tables do not show this.
Also, in the title authors notice “A comparison between institutionalized and general geriatric population" So, the aim needs correction!
Metods
How many respondents were involved in the follow-up? How many of them are included and how many are excluded? Number? How many were number institutionalized and how many numbers of the general geriatric population
How did the authors control the socio-demographic variables? Namely, is it possible that the respondents who were hospitalized in the hospital, for example, in 2016, still have the same conditions of life?
It is not entirely clear how the subjects were monitored. How did the authors find out these data, if the patients were not hospitalized again) Did they contact them by phone? Were there any who were lost in the follow-up (e.g. they died...) It is not clear what the data were at the beginning and what they were 24 months later
"This retrospective study was conducted .... between 2016-2020" What does this mean? The authors noticed that they followed the patients for 2 years. Does this mean that they patients followed for 4 years??? It is necessary to describe all this precisely because it is unclear
Results
The authors wrote: "The clinical and demographic data of the patients included in our study are presented in Table 1." Is there a difference between the three observed groups? It is quite unclear whether the groups differed in any way, and p-value is needed. Table it would be better to divide into visible columns for three groups: institutionalized patients, patients living alone, and patients living within the family. Accordingly, it is necessary to find differences between study groups and give p-value.
Table 2 is unreadable. What does the p value refer to? "Total" is completely redundant here. It should be clearly visible that the difference refers to groups in relation to "present or absent falls".
Table 3 is also unclear. It is not clear to whom OR refers. For example, what is the OR for Institutionalized and how much for Living alone, and so on for all compared squares? You should clearly state the OR for each group and then see the p-value.
In a note of the tables, a legend of the aberration in the table (for example, TKA, THA), is needed.
“Two of these cases are presented in Figures 1 and 2.” Are these figures relevant to the aim of the study? In my opinion, these pictures are unnecessary, because this article doesn't analyze of the surgical method.
Discussion
The discussion is confusing and needs revision. I miss the "red line". It would be improved if you start to describe the main results of your study and continue to associate it with other studies on the same topic. Thought all discussions the authors describe the theoretical and scientific concept of their own study topic but do not relate it to their results. The discussion should clearly explain WHY the results are just like that and what they mean
it is necessary to better connect your results with results from other studies and explain why.
In general, the article (especially the results and tables) is poorly conceived, despite the interesting and important topic.
In my opinion, this article has severe limitations and is not acceptable for publication in this journal
Author Response
Dear Reviewer,
Thank you for evaluating our manuscript and for your valuable comments. Below you may find our response to each of your observations. We hope we have managed to properly improve the manuscript.

Reviewer 2 Report
The paper is interesting and professionally written. Some point should be considered:
Abstract
· “… indicating the interest of nursing homes in offering 22 proper care.” This statement is subjective and not based on the findings of this study. Please delete it.
Introduction
· No comments
Materials and methods
· Inclusion criteria: authors included patients with indications for TJA following trauma or OA. Actually, they are two very different populations. Subjects that need TJR are usually ones who fractured the femoral neck after falling. It is known that previous falls is a predictor of falling. So, it is interesting to analyze these subsamples separately. If it cannot be done, please mention this in a limitation section.
· “falls reported in the next two years after TKA or THA” please acknowledge that falls reported in a medical history may be inaccurate because not all falls are reported to medical authorities.
· Please provide some information on the medical interventions that patients receive in nursing homes and in the community. In some countries, physical therapy is an integral part of nursing home services. Same in the community, patients after TKR and THR usually receive home-based rehabilitation and, when possible, are transferred to community physical therapy clinics. Is this the same situation in Romania? Without this information, it is difficult to interpret the study's results.
Results
· Table 2. Columns 4 and 7 (Total) are unnecessary and can be deleted.
· Figures 1 and 2 are unnecessary for understanding the results of this study and can be deleted.
· What I fill missing is an analysis of fallers vs. non-fallers (prior to TJR)
Discussion
· No comments
Author Response

(The authors gave the same response as above.)

Reviewer 3 Report
This is an interesting study about prevalence of fall after TKA and THA. Furthermore, authors tried to estimate prevalence of fall between three different living conditions. However there are several queries need to be resolved before reviewer give a decision.
1. This study only use chi square and fisher exact but no multivariate analysis was done. So we can't know for 100% if the effect of different living conditions is as mentioned in the article. Please conduct a multivariate analysis using logistic regression for both TKA and THA groups adjusting for covariates.
2. it will be interesting to know the difference characteristics between living conditions by stratified the result in table 1 into 3 different living conditions. Because, we can understand what is the difference between those 3 living conditions.
3. regarding re-intervention, is there any difference in characteristics between those with re-intervention and those without it?
Author Response

(The authors gave the same response as above.)

Reviewer 4 Report
The study examines the associations between living conditions and the prevalence of postoperative falls following joint replacement. A set of studies have investigated this issue. Therefore, I don’t see any significant contributions of this study. Additionally, authors are analyzing “living arrangement”, instead of physical “living conditions”. I would suggest replace terms and paragraphs in the literature review section. Last, this study does not deal with any heterogeneity issue (I.e. bi-directional causality and missing variable) and, therefore, cannot conclude that living conditions can influence the prevalence of falls but could only say they are associated.
Author Response

(The authors gave the same response as above.)

Round 2
Reviewer 1 Report
I have finished the second review.
The authors partially improved the article, but I still think that the work has shortcomings.
especially in tables.
for example, the bottom of the table is missing which statistical test was performed (Table 1 and Table 3). Why do authors take out % from table 2,
Besides, I don't know why is done in Table 3. post-hoc analysis based on Fisher's exact test for Odds Ratio???
in my opinion, this is a very unusual statistical procedure! Did the authors have some references for that procedure?
Odds radio has its own interpretation and significance, and you should not compare two Odds Ratios
"The odds ratio tells us how much higher the odds of exposure are among case-patients than among controls. An odds ratio of 1.0 (or close to 1.0) indicates that the odds of exposure among case-patients are the same as, or similar to, the odds of exposure among controls" or other comparison groups (in this case, exposure are falls)
The discussion is not improved, especially the first sentence "red line" and posibility of generalisation.
Author Response
Dear Reviewer,
In the attachment you may find our response to your comments.

Reviewer 3 Report
authors have revised according to reviewer's comments and now ready to be published.
Author Response

(The authors gave the same response as above.)
